# Prompting as Scientific Inquiry

**Ari Holtzman**       **Chenhao Tan**

**University of Chicago**
{aholtzman,chenhao}@uchicago.edu

## Abstract

Prompting is the primary method by which we study and control large language models. It is also one of the most powerful: nearly every major capability attributed to LLMs—few-shot learning, chain-of-thought, constitutional AI—was first unlocked through prompting. Yet prompting is rarely treated as science and is frequently frowned upon as alchemy. We argue that this is a category error. If we treat LLMs as a new kind of organism—complex, opaque, and trained rather than programmed—then prompting is not a workaround. It is behavioral science. Mechanistic interpretability peers into the neural substrate, prompting probes the model in its native interface: language. We argue that prompting is not inferior, but rather a key component in the science of LLMs.

## 1  Introduction

As Olah et al. (2020) makes the case for mechanistic interpretability, it argues that neural networks are an object of empirical investigation:

> In this view, neural networks are an object of empirical investigation, perhaps similar to an organism in biology. Such work would try to make empirical claims about a given network, which could be held to the standard of falsifiability.

We believe that the same reasoning applies to prompting. Prompting is not merely a way to use language models—it is a form of scientific investigation through language, the model's native interface. If we discovered an intelligent alien species, we would learn many things from playing simple card games and observing their reasoning patterns that we would have trouble learning by dissection. Similarly, prompting allows us to probe LLMs through their optimized communication channel, revealing capabilities and limitations that might remain hidden in their weights and activations if we don't know what to look for.

However, prompting is frequently considered inferior to other methods, especially those used in Mechanistic Interpretability that attempt to explain behavior at the level of neurons or individual parameters. When discussing methods for improving models, researchers commonly dismiss methods of structure prompting, treating it as a mere hack rather than genuine progress. Until, of course, such a method is endowed with a name and enough proven success so that it is viewd as distant from the word "prompting", e.g., DsPY (Khattab et al., 2024).

This cultural attitude stands contrary to reality: prompting remains the most impactful method we have for discovering what lies inside LLMs since the leap from language models to large language models. Training innovations have made these models better, more efficient, and capable of handling longer contexts, but it has generally been difficult to explore LLMs effectively through optimization methods alone. Instead, we have relied on the richness of language, with all the subproblems it can express, to prompt LLMs into revealing their capabilities and limitations.

39th Conference on Neural Information Processing Systems (NeurIPS 2025) Position Paper Track.

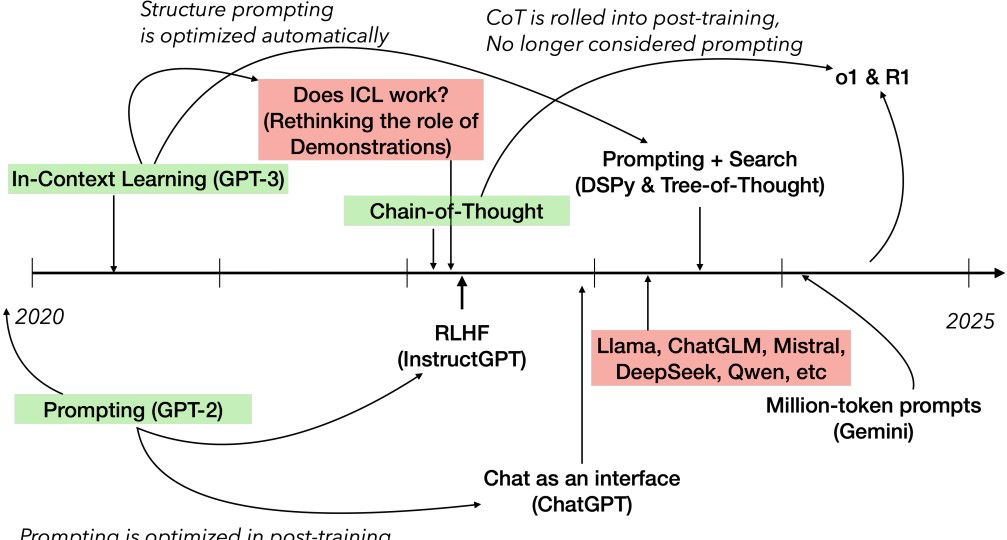

Figure 1: An illustration of the timeline related to the success of prompting (highlighted in green) and potential cause of the decline in the perception of prompting (highlighted in red).

The bias against prompting reveals our field's persistent preference for optimization over exploration, for mathematically elegant solutions over empirical discovery. Yet history repeatedly demonstrates that prompting has been the source of our most significant insights. In-Context Learning wasn't the projected outcome of scaling-up Transformers, it was a delightful side-effect that changed the game of what we could expect from model generalization at inference time. Chain-of-thought reasoning wasn't discovered through architecture innovation but by simply asking models to "think step by step." Many alignment techniques are, at their core, elaborate prompting methods dressed in mathematical formalism to appeal to the aesthetics of rigor in ML communities.

Behind the hand-waving dismissals lies an uncomfortable truth: prompting reveals model capabilities we never knew to look for, while interpretability has, so far, largely confirmed hypotheses we've already formulated. Prompting is a form of discovery, and it's badly needed to explore the complex and opaque new organism we have been confronted with: Large Language Models.

In this paper, we aim to advocate prompting as an important avenue of scientific inquiry. We start by highlight the impact that prompting has led to in the recent advances of large language models (Section 2), and then provide our account of how prompting got counted out (Section 3). We argue that prompting should be a critical form of scientific inquiry for interrogating LLMs (Section 4) by 1) providing a direct comparison between prompting and mechanistic interpretability to illustrate that they are complementary and critical methods to understand and control large language models (Section 5), and 2) giving direct rebuttal to counterarguments in Section 6.[1] Finally, we lay out concrete future directions to pursue prompting as science (Section 7).

## 2 Why Defend Prompting? Three Case Studies

Prompting is not just a way to use models, it is the primary means by which we have discovered what they can do. Across model generations, discovery via prompting has consistently outpaced our understanding of the internal structure of LLMs, surfacing capabilities that would have otherwise gone unnoticed. Importantly, these discoveries were rarely the result of theoretical foresight or interpretability breakthroughs. They came from structured interaction: trying things out, observing behavior, and refining prompts until something new emerged. Many people would describe this as unscientific. The problem is that this is the essence of scientific discovery—intervening on key

---

[1]To make the argument non-trivial, we focus on the case of understanding, developing, and controlling LLMs. It is already very clear that prompting is the essential ingredient in the emerging area of AI agents.

variables in a system is how we discover new effects to test in the first place, it just so happens that the inputs to the this system is human language.

Below, we present three case studies of this dynamic. Each represents a major leap in how we understand and control LLMs—one that began not with weight analysis, but with well-chosen words.

**Case Study: Sparks and Embers.** An essential issue that researchers must grapple with regarding LLMs is the demand to measure what new capabilities models may have, by finding instances of novel behavior. Claims of novel behavior often get debunked when behavior does generalize as one might hope—just because a model can solve 1 digit addition problems does not mean it can solve three digit ones—an issue that psychologists run into as well (Yarkoni, 2022). This statement and response recently happened in the conversation between "Sparks of Artificial Generative Intelligence" (Bubeck et al., 2023) and "Embers of Autoregression" (McCoy et al., 2024). Bubeck et al. (2023) is an exploratory study of what an early version of GPT-4, which suggested a number of new capabilities such as the ability to generate stories with embedded hidden messages (via acrostics) or write a small 3D game from scratch. How were these discovered? Through prompting. Yet McCoy et al. (2024) showed some of these incredible abilities didn't generalize in the way we'd hope, e.g., that the model can *decode* a shift cipher[2] but have trouble *encoding* that same cipher. Similarly to Bubeck et al. (2023), this was shown through careful prompting studies, because this is the most natural interface under which to manipulate the conditional distribution of LLMs.

**Case Study: Chain of Thought.** Wei et al. (2022) showed that asking models to generate human-like reasoning chains actually resulted in better reasoning outcomes (Figure 1), building on the tradition of prompting to discover behavior that popularized by GPT-2's prompting studies, e.g., using "TL;DR" for generating summaries (Radford et al., 2019). The intuition was to find situations where humans might generate such reasoning chains and *prompting* models to enact this behavior. The key insight wasn't that models simply needed more raw computational power and engineering that into a model manually, though later studies showed that models could use inference-time compute without "thinking" out loud (Pfau et al., 2024). Researchers wanted to take advantage of behavioral patterns models may have learned from their training corpus, using the correlations between reasoning chains and correct outputs. This discovery has since spawned a new generation of training procedures that leverage *inference-time compute* strategies. Modern systems like DeepSeek-R1 (Guo et al., 2025) and GPT-4o1 (OpenAI, 2024) now demonstrate that models can reason better in recognizable ways—achieving improved results on complex mathematical problems and challenging domain-specific tasks such as graduate-level questions in GPQA (Rein et al.). Asking models to show their work revealed how we could train models to do something more like thinking—a process that would have been difficult to achieve with anything but prompting. While it's clear that prompting led to genuine modeling developments, it was nonetheless prompting that led to these developments.

**Case Study: Constitutional AI as a method of alignment and control.** Prompting is also the backbone of many alignment strategies. In Constitutional AI (Bai et al., 2022), prompting is used not only to elicit desirable behavior, but to enforce constraints and shape it. The process works by having models generate self-critiques and revisions, then fine-tuning the original model on these revised responses. Rather than rely entirely on human feedback, models are prompted to critique their own answers against a predefined set of normative principles (the "constitution"), and then to revise those answers accordingly. Human oversight is provided through a list of rules or principles expressed in natural language—it is this form of natural language prompting that governs the entire process. This method doesn't modify the model's internals—it prompts itself into generating training data that aligns the model. Here prompting serves as both a tool of supervision and instrument of control. Many researchers have been skeptical that models could improve by training on their own outputs, believing these systems should have "nothing more to give." Like Chain-of-Thought, this strategy was not born from theoretical introspection or circuit-level understanding, but from treating the model as a behavioral agent that could be instructed, guided, and corrected through language.

## 3 How did Prompting Get Counted Out?

Given the huge impacting prompting has had on the evolution, use, and theory of LLMs, how did it become "dark magic"? We provide our interpretation of what has happened.

---

[2]A shift cipher is one where the alphabet is treated like a ring and letters are "shifted" by a fixed amount, e.g., in Rot3 letters are shifted by 3, so $A \rightarrow D, B \rightarrow E, \cdots, Z \rightarrow C$.

First, a series of studies demonstrated the sensitivity of prompting (Arora et al., 2022; Sclar et al., 2023; Wallace et al., 2019, *inter alia*). Model behavior can be unpredictable, counterintuitive, and small changes can change the performance of a system drastically. As a result, prompting cannot be good, and many researchers have suggested we need to find a way to get rid of it (Kosch & Feger, 2025; Meincke et al., 2025; Morris, 2024, *inter alia*).

Second, there is a strong bias in the machine learning community towards algorithms, modeling, and training methods (Birhane et al., 2022; Lipton & Steinhardt, 2019; Sculley et al., 2014, *inter alia*). This bias, combined with the increasing availability of open-weight models, caused researchers and practitioners to continually pursue bespoke systems with novel methods when large models consistently outperformed them with proper prompting (Nori et al., 2023).

Consequently, a curious pattern has emerged: the moment prompting discoveries proved transformative, they were quietly rebranded and distanced from their origins, see Figure 1. Chain-of-Thought reasoning (Wei et al., 2022) was discovered via In-Context Learning and then simplified to only require four words: "Let's think step by step" (Kojima et al., 2022). Once this proved effective, it became "inference-time compute" and was rolled directly into training procedures for models like o1 and DeepSeek-R1 (Guo et al., 2025)—no longer considered "prompting." RLHF (Ouyang et al., 2022) and ChatGPT (OpenAI, 2022) essentially automated effective prompting strategies into post-training, but were celebrated as training innovations rather than systematized prompting.

The pattern continued across multiple fronts: structured prompt optimization in DSPy (Khattab et al., 2024) is touted as "programming—not prompting—Foundation Models", while Tree-of-Thought (Yao et al., 2023) framed itself as "exploration over coherent units of text" rather than self-prompting. Constitutional AI (Bai et al., 2022) and Self-Instruct (Wang et al., 2023) used prompting for data augmentation, but were positioned as training methodologies that removed the variance of prompting, which was implied to be manual and unreliable.

Meanwhile, the field developed longer contexts—Gemini's million-token capacity (Team et al., 2024) enabling many-shot prompting (Anil et al., 2024)—precisely to accommodate more sophisticated prompting strategies. Yet these advances were celebrated as architectural achievements, even though *the driving motivation behind long-context models is to be able to prompt with more data*. However, we have more and more long-context training papers, but not a lot of exploration on prompting strategies with long context models. The uncomfortable truth remains: virtually every major capability breakthrough originated from careful prompting experiments, yet prompting itself continues to be dismissed as unscientific hackery.

## 4 Prompting as Scientific Inquiry

In this work, we define *prompting* as interacting with LLMs with natural language and observing their behavior, including the output texts and/or the probability distributions. Prompting is useful for two reasons that mutually support each other: (1) humans have a deep understanding of the structure of language and (2) since language models were discovered not designed (Holtzman et al., 2023), prompting is the most direct way to discover novel behavior in LLMs. This makes prompting a crucial form of scientific inquiry for interrogating LLMs and the structure of their computations.

Other methods we have available (e.g., discovering locally linear decision boundaries, statistical testing, etc.) largely select hypotheses from a very constrained set or reject null hypotheses. What we want, is a way to discover new capabilities, without having to be overly prescriptive about the hypothesis set. More formally, we want to find behavior in LLMs that was *a priori* unlikely (because previous models couldn't do it), but *a posteriori* predictable (because it can be used in reliable ways, e.g., ICL or CoT).

Despite the fact that all LLM researchers prompt models to understand aspects of how they work, it has become mentally associated with prompt engineers attempting to find tricks that improve benchmark numbers but are not generalizable. Part of this is the low barrier to entry, since many who use prompting do not use it rigorously. Yet this is hardly an argument against using it rigorously.

Putting LLMs in new contexts, examining the generated outcomes, and attempting to find patterns in the resulting probability distributions, these are precisely the ingredients of exploratory science when we are confronted with a new system that evades explanation. The exact same point was made in eight years ago in Neuroscience, by a paper titled "Could a neuroscientist understand a microprocessor?"

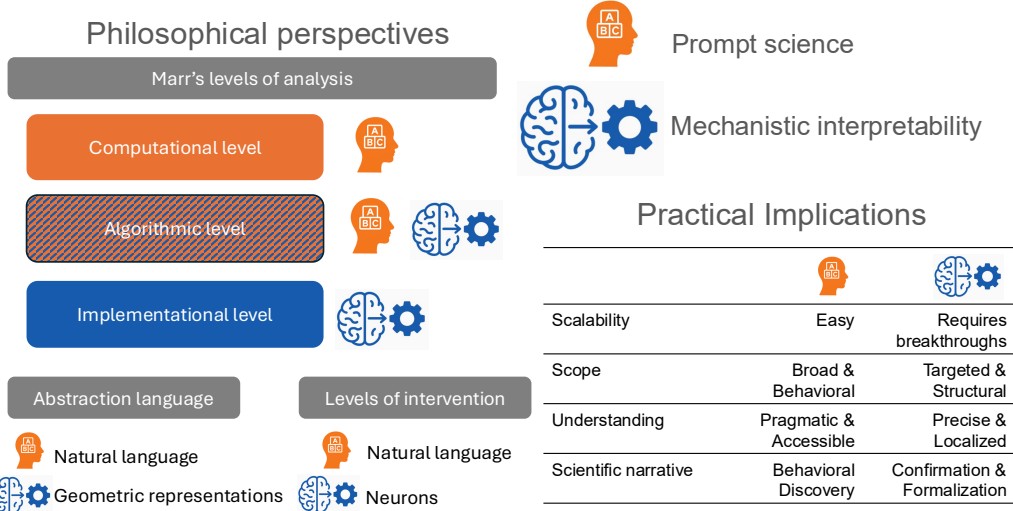

Figure 2: An overview of the comparison between prompting and interpretability.

(Jonas & Kording, 2017). The answer is "No." precisely because the strategies taken would not exploit the regularity with which these systems could be manipulated. LLMs have been handed to us on a silver platter, from this perspective: they already speak *approximately* the same language we do, shouldn't we be excited to be able to probe them with it?

Contrast this with a recently popular line of work in mechanistic interpretability: of Sparse Autoencoders (SAEs) (Bricken et al., 2023; Cunningham et al., 2024; Gao et al., 2024, *inter alia*). What is the goal of SAEs? It is to discover concepts in models we did not know were there. This is very similar problem to the one prompting solves, but there is a catch: SAEs have been popular, despite the plethora of evidence that they are unreliable (Makelov et al., 2024; Paulo & Belrose, 2025; Smith et al., 2025).

Why? Because of the *mythos of mechanistic interpretability*, which suggests that if we work with the internal mechanisms of models our results are not subjective like they are with language. Seven years ago Lipton (2018) explained that "claims regarding interpretability of various models may exhibit a quasi-scientific character" when the underlying concept remains ill-defined, warning us that the absence of clear definitions can create a false sense of scientific rigor where none actually exists. In the age of *mechanistic* interpretability we face a new but subtler issue: mechanistic interpretability has found ways of being more falsifiable than many previous interpretability methods (e.g., through interventional studies), but also **mechanistic interpretability has codified the idea that a rigorous statement about an LLM must be about model weights**.

This is simply bias against blackbox analysis. The probability distributions emitted by LLMs are just as much an objective part of their mechanistic description as anything else. The main source of error it introduces is human assumption: humans may think a comma in a prompt is meaningless, but the model may have noticed that in the training set a correctly placed comma achieves right answers to math problems 80% of the time, but an incorrectly placed one only 40%. This bias exists in mechanistic interpretability too, but is often subtler, e.g., the mistaken case of worrying about anisotropy in the embedding space described in Holtzman et al. (2023).

We do not at all believe that prompting and mechanistic interpretability are at odds with each other, but we do believe that prompting has been denigrated as a form of scientific inquiry to understand what mechanisms LLMs use. We therefore proceed in comparing the goals and methods of what we call "prompting" and the current state of mechanistic interpretability.

# 5 Prompting vs. Mechanistic Interpretability

To compare prompting with mechanistic interpretability, we start with three philosophical perspectives, and then discuss their practical implications.

## 5.1 Philosophical Perspectives

**Different levels of analysis.**    We start with a perspective inspired by Griffiths et al. (2024). David Marr's levels of analysis provide a compelling framework for understanding how prompting and mechanistic interpretability complement each other (Marr, 1982). Marr proposed three levels for understanding complex information-processing systems:

1. *Computational level*: What is the system doing and why?
2. *Algorithmic level*: How does the system do it, in terms of representations and processes?
3. *Implementation level*: How is the system physically realized?

Prompting excels at revealing insights at the computational level—discovering *what* capabilities models have and *why* they might have developed them through training. When we discover that LLMs can perform few-shot learning through careful prompting, we are gaining computational-level insights about the model's function. When we find what kind of prompts elicit certain behavior, we can often get a sense of what training data these capabilities were influenced by—giving us a peek into the why. For instance, McCoy et al. (2024) show that the model can use shift ciphers with shift values that are commonly used on the internet (1, 3, and 13) but have drastically reduced performance on other values. In comparison, mechanistic interpretability shines at the implementation level, revealing how specific patterns of weights and activations realize particular functions. For instance, mechanistic interpretability researchers used the structure of modulo arithmetic to examine generalization Liu et al. (2022) and eventually Nanda et al. (2023) fully reverse engineering the algorithm used in small transformers trained on modular addition tasks, which uses discrete Fourier transforms and trigonometric identities to convert addition to rotation about a circle. This led to studies showing that multiple different strategies for modulo arithmetic can be learned Zhong et al. (2023), and eventually scaled-up studies examining these in medium-sized LLMs (Kantamneni & Tegmark, 2025).

Both approaches meet at the algorithmic level, where we aim to understand the representations and processes models use. Prompting can reveal algorithmic insights through structured behavioral probes, while mechanistic work can in principle identify the specific computational graphs that implement these algorithms. Prompting is better at helping us map out the diversity of possible behavior in many different contexts and is easy to use on models of any scale, where as mechanistic interpretability shines when attempting to identify small differences in processing and often requires many papers to scale-up.

**Different abstraction languages.**    A fundamental distinction between prompting and mechanistic interpretability lies in their abstraction languages. Mechanistic interpretability relies on geometric abstractions in representation space—circuits, feature directions, and activation patterns. These offer precision but at the cost of having to fully specify definitions, in a way that can limit their applicability. Prompting, by contrast, leverages linguistic abstractions through the manifold of language itself. Importantly, since LLMs are trained on such huge linguistic corpora, we should expect the structure of language to mirror aspects of the model itself.

Language-based abstractions enable what we might call "productive vagueness"—they can specify hypotheses at varying levels of precision, matching our own incomplete understanding of complex phenomena. When we prompt a model to "think step by step," we are not specifying a precise geometric direction in activation space, but rather invoking a rich cluster of associations that the model has learned. This resulting chain-of-thought allows models to expose aspects of what they are doing that would be difficult to describe in more mechanistic vocabulary.

Crucially, linguistic abstractions naturally align with human conceptual structures, making prompting discoveries more interpretable and actionable for both researchers and practitioners. The tradeoff is precision—linguistic prompts allow a certain ambiguity that mechanistic approaches aim to eliminate. We can easily fool ourselves into thinking we understand more than we do about models, by mixing up our understanding of words with theirs. Yet this ambiguity often mirrors the distributed, entangled nature of how concepts are actually represented in neural networks—and strong departures from human semantics provide a powerful signal to where faithfulness will be hard to achieve.

**Different levels of intervention.**    Both prompting and mechanistic interpretability can be understood as different approaches to intervention-based science. Mechanistic interpretability intervenes on

internal model components—modifying weights, freezing activations, or ablating attention heads—to establish causal relationships between components and behaviors.

Prompting intervenes at the model's natural interface—the input distribution it was trained on—to similarly establish causal relationships between inputs and behaviors. Both are forms of causal intervention, just at different levels of the system.

This perspective helps explain why prompting has been more immediately productive: it leverages the model's native interface, which is already optimized to produce meaningful responses to linguistic inputs. The model's architecture has been specifically designed and trained to transform language inputs into language outputs. Prompting takes advantage of this by using the interface the model was built to understand.

## 5.2  Practical Implications

Next, we discuss the practical implications of the above differences along five dimensions: scalability, scope, when mechanistic interpretability excels, understanding, and scientific narrative.

**Scalability.** Mechanistic interpretability struggles to scale with the complexity of modern LLMs. Reverse-engineering billions of parameters is costly if not impossible, and it's unclear whether the resulting explanations are actionable at scale (Holtzman et al., 2023; Räuker et al., 2023). Prompting, by contrast, is lightweight and deployable at scale, enabling efficient behavioral testing and control without circuit-level access.

**Scope.** Mechanistic interpretability requires precise hypotheses about model internals, which is often infeasible, especially when the relevant features or circuits are unknown. Moreover, composing multiple edits or identifying the right granularity remains an open challenge. As a result, studies have so far focused on explaining and controlling simple and constrained behavior. Prompting operates at the level of language, giving it broad expressive power to specify desired behaviors or test for failure modes, including subtle or complex ones (Garbacea & Tan, 2025; Rajani et al., 2023). However, prompting has important limitations. It requires models with sufficient linguistic competence to meaningfully respond to natural language instructions. Small models or those trained on limited data may not exhibit behaviors discoverable through prompting. Pre-linguistic models—those that lack basic language understanding—cannot be studied via prompting at all. Additionally, tasks requiring precise sub-token manipulation or exact algorithmic control may be better suited to mechanistic approaches that operate below the level of language.

**When Mechanistic Interpretability Excels.** Despite these limitations, there are scenarios where mechanistic interpretability is clearly the better tool. When research requires precise algorithmic reverse-engineering—such as identifying the exact Fourier transform circuits used for modular arithmetic (Nanda et al., 2023)—mechanistic methods provide unmatched precision. Similarly, understanding specific architectural components, investigating computation at granularities below the level of language representation (such as sub-token processing), and analyzing small models where full circuit tracing is feasible all favor mechanistic approaches. The key is matching the method to the research question: implementation-level questions demand implementation-level tools.

**Understanding.** Mechanistic explanations prioritize faithfulness, but their precision can make them opaque—like reading a circuit diagram. The over-emphasis on faithfulness loses track of its pragmatic utility. After all, the only truly "faithful" explanation is the computational graph for a given instance, but we all agree that such a faithful explanation is useless. By contrast, prompting generates behavioral evidence and accessible explanations, serving as a starting point for intuition and hypothesis generation. Prior work has emphasized that explanations are diverse and often shallow (Keil, 2006; Lombrozo, 2006; Tan, 2021; Wilson & Keil, 1998), suggesting a pragmatic role for prompting-based understanding.

**Scientific Narrative.** Many key discoveries in interpretability begin with behavioral anomalies. For instance, the characterization of induction heads followed observations of copying behavior in Transformers (Al-Rfou et al., 2019; Elhage et al., 2021). Prompting offers a structured hypothesis space to surface such behaviors, guiding mechanistic follow-up. In this light, prompting is not just an evaluation tool; it drives scientific discovery by revealing what is worth interpreting.

In summary, prompting offers both practical advantages and valuable insights comparable to mechanistic interpretability. Rather than competing approaches, they represent complementary methods—

analogous to how behavioral psychology and cognitive neuroscience both contribute to understanding human cognition.

# 6 Addressing Arguments for Dismissing Prompting

There are many arguments for dismissing prompting, we respond to those we think are most frequent and impactful below.

**Lack of mathematical formalism.** A common critique is that prompting lacks the mathematical rigor expected in machine learning research. The absence of clean equations to describe input-output relationships can make prompting feel less scientifically grounded compared to approaches with elegant mathematical formulations. However, this critique assumes that the most important phenomena in LLMs are best captured through mathematical abstractions we currently have notations and definitions for. As argued in §5.1, the abstractions prompting allows us to use to probe models are more natural for understanding model behavior than those we know how to capture in equations. A large part of the point of prompting studies is to discover how we can formalize these: In-Context Learning's language of input/output demonstrations, Chain-of-Thought's use of an internal monologue, and DSPy's method of turning prompt-optimization into programming. The equations we write today—whether describing attention mechanisms or optimization landscapes—often fail to capture the rich behavioral patterns that prompting reveals. Just as plant breeders developed systematic methods for eliciting desired behaviors in crops long before understanding the genetic mechanisms underlying those traits, prompting provides a framework for investigating LLM capabilities that may not yet be amenable to compact mathematical description until we discover "the genetic code" of how models interpret language.

**Lack of model training.** Machine learning culture biased towards the belief that meaningful research contributions must involve training new models or developing novel architectures (Birhane et al., 2022; Lipton & Steinhardt, 2019; Sculley et al., 2014, *inter alia*). This perspective can make prompting-based discoveries feel less substantial, as they don't require the computational resources or optimization-expertise associated with model development. However, this view conflates the prestige of model training with scientific value. Many of the most important insights about LLMs—from in-context learning to chain-of-thought reasoning—were discovered through careful prompting rather than architectural innovation. The field's reward structure may inadvertently discourage researchers from pursuing behavioral investigations that could yield fundamental insights about these systems, but that's hardly a good reason to dismiss the scientific contributions prompting has and can give us.

**Prompt hacking.** Prompting allows for a great deal of variability in how LLMs are deployed (Meincke et al., 2025), creating issues similar to p-hacking (Kosch & Feger, 2025) where one can optimize for the correct prompt to get betters scores on an evaluation. Critics argue that this variability should be eliminated to make systems operate like traditional computer programs (Morris, 2024). This represents a fundamental category error: it assumes we want to evaluate the LLM "in isolation" with the prompt getting in the way of that assessment. In reality, LLMs are probability distributions that do not yield answers to questions; only LLMs combined with prompts and decoding methods constitute complete systems that can be evaluated. Attempting to eliminate this variability through universal prompts misses the point entirely. LLMs naturally require context, and prompting is how we provide that context. **The real issue is insufficient prompt optimization.** If a better prompt can give better results, then that is capability of the system that "runs" on the given prompt, just like how we evaluate program efficiency on specific hardware, not with the program or the hardware alone.

**Prompts are brittle.** Extensive research has documented "prompt brittleness"—the fact that minor perturbations to prompts can dramatically change LLM outputs (Arora et al., 2022; Sclar et al., 2023; Wallace et al., 2019, *inter alia*). Often this is unintentional, and models fail to do the desired task after small, seemingly meaningless changes. While this certainly complicates model control from an end-user perspective, we argue that this sensitivity actually reflects models' attempts to infer substantial information from limited context. When LLMs lack sufficient context (from their perspective), they compensate through sensitivity to pragmatic implications, a consequence of training objectives that cause them to model implicit aspects of authorship and persona (Andreas, 2022). Instruction-tuning methods have reduced sensitivity to perturbations (Ouyang et al., 2022) but have also measurably reduced flexibility (Lin et al., 2024). This trade-off highlights that prompt sensitivity is a tool for exploring what factors influence model behavior. **Prompt brittleness may make prompt**

**engineering more difficult, but it makes the science of prompting significantly more potent.** Understanding how human intentions in prompts don't perfectly transfer to model outputs is a key mechanism for cataloging what models actually know, rather than merely an error to be corrected.

**Prompts will not generalize across models.**   Researchers often worry that prompting discoveries will be too model-specific to provide general insights about LLM behavior. This concern reflects valid scientific instincts about the importance of reproducibility and generalization. However, in practice, many fundamental prompting methods, including in-context learning, retrieval-augmented generation, and chain-of-thought reasoning, have proven robust across model families at large enough scales. Rather than being a weakness, the ability to test generalization across models provides a valuable filter for distinguishing fundamental capabilities from model-specific artifacts.

**Prompting requires excessive manual effort.**   While less explicitly discussed in academic literature, a significant barrier to taking prompting seriously as a research method is the perception that it is fundamentally more labor-intensive and less systematic than other approaches for studying LLMs. This is in contrast to the simultaneous but opposite belief that prompting is "trivial" because it simply involves writing text and observing model outputs. The field has grown accustomed to research methods that can be formulated as optimization problems with clear objective functions, positive examples for fine-tuning, reward signals for reinforcement learning, or compositional operations like task arithmetic (Ilharco et al., 2022). Prompting resists these familiar optimization frameworks; while it can technically be viewed as discrete optimization, the irregular and very incomplete understanding of the structure of language makes traditional optimization methods useful only in limited cases. Yet this perception of difficulty is often misleading. In practice, prompting frequently enables faster iteration cycles and cheaper exploration compared to methods requiring model retraining, making it an efficient tool for behavioral investigation rather than a cumbersome alternative. This is a difficult statement to prove directly, so we instead offer as evidence all the breakthroughs that prompting has led to, some of which are shown in Figure 1.

## 7   Future Directions for Making Prompting a Science

In this section, we lay out some concrete future directions for making prompting a scientific inquiry.

**Prompting as Mechanism Discovery.** We believe that prompting will continue to be an effective way to discover LLM capabilities and limitations. Developing a symbiosis between prompting and mechanistic interpretability will be critical for pushing the frontier of LLMs. We envision complementary workflows: prompting discovers ordered anomalies and behavioral patterns, mechanistic interpretability reverse-engineers the implementing circuits and validates causal hypotheses, and these mechanistic insights suggest new prompting experiments to test predictions. For instance, prompting might reveal unexpected reasoning capabilities, mechanistic analysis could identify the attention patterns responsible, and follow-up prompts could systematically test the boundaries of these patterns. This bidirectional cycle—from behavior to mechanism and back—mirrors successful paradigms in neuroscience and psychology. Example questions include

- Can we use systematic prompt perturbations to map the precise boundaries where model capabilities break down, revealing the implicit structure of learned knowledge?
- Can we find concepts that only machines, not humans, use (Hewitt et al., 2025; Holtzman et al., 2023)? How can we communicate such concepts to humans?
- Can we use controlled prompt interventions to discover causal dependencies between model capabilities, revealing how different types of knowledge and reasoning interact within LLMs and providing hypotheses for mechanistic interpretability?

**Prompting for Effective Control of LLMs.** Building on the above insights, prompting can be a productive tool for enabling effective control of LLMs. However, much work is required to make this a science. Example questions include

- Can we identify a minimal set of "control primitives" in prompt space that can be composed to achieve arbitrary behavioral modifications, analogous to how universal computation emerges from simple logical operations? What are the dimensions of control that are and aren't possible?

- Most prompting strategies have been constrained within a single long prompt and recent studies show that this is actually more effective than multi-turn prompting (Laban et al., 2025). What is special about long-prompts? What information do multi-turn prompts fail to capture? What makes effective strategies in a multi-turn setting?
- Can we develop adversarially robust prompting strategies that maintain control even when models encounter inputs designed to subvert intended behavior, leveraging multi-model verification (e.g., "is this request harmful?"), explicit reasoning chains, and meta-cognitive prompting?

**Prompting as Future Proofing.** Whatever future models may look like in architecture, modality, training, or hardware—we will certainly use text to help establish what we want. Focusing on prompting can help us future proof this process. Example questions include

- What prompting strategies are robust to model updates? What defines a "forward-compatible" prompt? And vice versa, how can we develop models that are backward-compatible with prompts? Is this a way to disambiguate natural language instructions for people, not just LLMs?
- Can we use patterns discovered through current prompting research to predict and engineer the behavioral properties of future AI systems, essentially creating a "prompting-informed design principles" framework that shapes how next-generation models should be trained?
- Can we identify universal principles of inter-intelligence communication through prompting that will remain valid even as AI systems become vastly more capable than humans—essentially discovering the fundamental "protocol" of how minds communicate across intelligence gaps?

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
