# OpenReview forum: "Prompting as Scientific Inquiry"
_NeurIPS.cc/2025/Position_Paper_Track — NeurIPS 2025 Position Paper Track_

### Official Review · Reviewer_yFki · 2025-07-19

**Significance:** 3
**Presentation:** 2
**Rating:** 6
**Confidence:** 5

**Summary:**

This paper argues that prompting should be taken seriously as a scientific methodology for interrogating and understanding large language models (LLMs). Drawing on analogies from biology and behavioral science, the authors suggest that prompting, structured interactions using natural language, is akin to behavioral experimentation on a new type of opaque, emergent system. Through historical and contemporary examples (e.g., chain-of-thought, in-context learning, Constitutional AI), the paper demonstrates that prompting has enabled nearly all major breakthroughs in LLM capabilities. The authors critique the ML community’s preference for internal, mechanistic explanations and formal training innovations, showing that prompting offers a complementary and often more practical pathway for insight. They argue that prompting is scalable, flexible, and under-theorized, and lay out a roadmap for treating it as a rigorous scientific inquiry alongside mechanistic interpretability.

**Strengths:**

The paper clearly articulates an underappreciated but crucial perspective: prompting is not a fringe technique or mere engineering artifact, but a foundational method of scientific discovery in the era of LLMs. The authors support their position with strong case studies, a philosophical framework (Marr’s levels), and practical implications. It bridges conceptual and practical divides, offering a roadmap for future work. The style is confident, grounded in evidence, and highly readable.

**Weaknesses:**

While the argument is compelling, it is largely discursive and could benefit from more empirical support (e.g., metrics showing impact of prompting-based discoveries over time). Additionally, it primarily focuses on prompting in LLMs and may not generalize as easily to other subfields of ML (e.g., vision). The paper might also be improved by deeper discussion of how to ensure rigor in prompting research (e.g., reproducibility standards, benchmark design). Finally, while the comparison with mechanistic interpretability is thoughtful, it may come off as too critical in places, potentially limiting its persuasive reach.

**Questions:**

- What methodological frameworks would you propose for making prompting research more systematic and reproducible?
- Could prompting be unified with interpretability through hybrid tools or evaluation pipelines?
- How do you see prompting evolving as models become multimodal or increasingly autonomous?

**Alternative Position:**

Yes, and alternative positions are trivial straw-man arguments

**Author Identification:**

No.

**Context:**

2

**Discussion:**

3

**Ethics:**

["NO or VERY MINOR ethics concerns only"]

**Position:**

Yes, the paper argues for or against a position related to machine learning.

**Support:**

3

**Thoroughness:**

5

---

### Official Review · Reviewer_ApcN · 2025-07-29

**Significance:** 3
**Presentation:** 3
**Rating:** 7
**Confidence:** 4

**Summary:**

Authors present a position to uplift the tarnished image of "prompting" from being considered a hacky non-scientific method to increase model performance to an actual method for scientific enquiry in and using LLMs.

**Strengths:**

- I really enjoyed reading this paper. It takes a simple point of view, a position, and argues for it quite thoroughly.
- It presents case studies and examples on why the alternate positions (e.g., studying LLMs through mechanistic interpretability) do not originate solely from merit but convention. The convention of feeling more "scientific" with mathematical equations, etc.
- It presents several potential arguments against their position and defends them.
- Finally, it also presents several future avenues of research if one was to agree with their position.
- In general, this is a good example of how a position paper should look like, in my opinion.

**Weaknesses:**

- The paper's position fails to acknowledge that prompting only works for models trained on natural language. Say we build a hypothesis in LLMs through mechanistic interpretability. We can test the same hypothesis on synthetic languages on very small scale transformer models to see what scale/data properties are necessary for a particular behavior to show up. Prompting will only work for models of reasonably large scale which can use the natural language interface effectively.

**Questions:**

- How would you use prompting to study ineffective models, or not completely trained models, models that have not [yet] developed the ability to use the natural language interface effectively?

**Alternative Position:**

Yes, and alternative positions are well-considered and addressed by the argument

**Author Identification:**

No.

**Context:**

3

**Discussion:**

3

**Ethics:**

["NO or VERY MINOR ethics concerns only"]

**Position:**

Yes, the paper argues for or against a position related to machine learning.

**Support:**

3

**Thoroughness:**

4

---

### Official Review · Reviewer_omDN · 2025-08-14

**Significance:** 3
**Presentation:** 3
**Rating:** 7
**Confidence:** 4

**Summary:**

The paper argues that prompting should be treated as a scientific inquiry into LLMs, akin to behavioral science, rather than “alchemy.” Prompting probes models in their native interface (language) and complements mechanistic interpretability, which inspects internals. It advocates that prompting is a primary, scalable, and rigorous route to discover and control LLM behavior. The contributions it made to support such a position include:
1. Synthesizes evidence that prompting unlocked major capabilities (few-shot, chain-of-thought, constitutional AI), illustrated via three case studies: “Sparks vs. Embers,” CoT leading to inference-time compute, and Constitutional AI as prompt-driven alignment.
2. Diagnoses why prompting is devalued: sensitivity/brittleness results, community bias toward training/algorithms, and rebranding prompt discoveries, etc
3. Proposes a comparative framework using Marr’s levels (computational/algorithmic/implementation), contrasting linguistic vs geometric abstraction languages, and input-level vs internal interventions.
4. Rebuts common critiques (lack of formalism, “prompt-hacking,” brittleness, poor cross-model generalization, manual effort).
5. Outlines a research agenda with several directions

**Strengths:**

1. The paper clearly and persuasively argues for prompting as rigorous scientific inquiry, provoking valuable discussions.
2. It deeply addresses why prompting is undervalued, using historical context, community biases, and Marr's levels to illustrate its complementary role alongside mechanistic probing.
3. The paper thoughtfully addresses common critiques of prompting, effectively resolving potential reader concerns and reinforcing the relevance of the topic to the NeurIPS community.

**Weaknesses:**

1. The position is argued largely through rhetoric and analogies. Claims about prompting’s effectiveness, generalizability, and scientific status might need to be backed by controlled studies, predictive theory, or deep engagement with Marr’s levels. The paper can better convincingly show what new, concrete gains we get by treating prompting as a formal scientific inquiry (e.g., case studies where prompting yields insights/measurable advances unavailable to baselines or MP).
2. The paper should also discuss: observations from prompting are hard to formalize/summarize, which impedes the accumulation of knowledge. The complementary role of prompting with mechanistic interpretability should be more formal and articulated.
3. If LLMs are treated as organism-like complex systems, the paper should import methods from behavioral/neuroscience (controlled stimuli, counterbalancing, preregistration, power analyses, cross-model generalization). It also sidesteps the core dilemma: prompting feels easy, but rigorous prompting research is hard. The paper should delineate standards that separate casual prompting from serious inquiry (experimental design, ablations, negative results, documentation) to justify the proposed scientific status.

**Questions:**

I enjoy reading this paper (the same fun as reading a thought-provoking blog), and I wonder the following questions:
1. If we consider LLMs as complex systems that happen to use language to communicate with the outer world, what kind of theories/analyses/etc can we borrow from neuroscience, cognitive science, or psychology
2. Since the language will be the main communication medium with this complex system, I wonder whether we need to reinvent linguistics (prev linguistics is for human-human communication)
3. If we want to treat "prompting" differently as a scientific inquiry, we may need to have a different name (same as how other useful/successful methods are rebranded). Simplyn using the name of“Prompting” isn’t crisply delimited, hindering it from becoming a distinct research object

**Alternative Position:**

Yes, and alternative positions are trivial straw-man arguments

**Author Identification:**

No.

**Context:**

3

**Discussion:**

3

**Ethics:**

["NO or VERY MINOR ethics concerns only"]

**Position:**

Yes, the paper argues for or against a position related to machine learning.

**Support:**

2

**Thoroughness:**

4

---

### Note · Authors · 2025-09-03

**1-10 Additional Comments:**

The track was valuable. A more detailed rubric would improve reviewer consistency, especially since reviewers sometimes asked for non–position-paper content.

**1-11 Submit Again:**

Definitely yes

**1-1 Submission Process:**

5

**1-2 Next Year:**

Keep a dedicated Position Paper Track with a rubric focused on clarity of argument and evidence, not technical novelty.

**1-3 Future Development:**

Publish a clear reviewer rubric and encourage more consistent evaluation, and focus on how useful a paper would be to discussions rather than the claims it makes. That's what position papers are ultimately about: moving the discussion forward!

**1-4 Interest:**

["Panel discussions with other position paper authors", "Structured debates on controversial topics"]

**1-5 Thoughtful:**

7

**1-6 Supportive:**

7

**1-7 Technical Aspects Versus Position:**

4

**1-8 Gate Keeping:**

9

**1-9 Camera Ready Changes:**

We will:

- Surface the three case studies more prominently and point to the timeline figure when referenced in text (Sparks vs. Embers; Chain-of-Thought; Constitutional AI).
- Tighten the Marr’s-levels signposting so each case study is clearly linked to its level(s) of analysis.
- Make the scope limits explicit (e.g., small or pre-linguistic models) and indicate when mechanistic methods are the better tool.
- Emphasize complementarity with mechanistic interpretability.
- Add editorial polish for titles, cross-references, and flow.

**3-1 Review Response1:**

omDN

**3-2 Reaction To Review1:**

Thank you for a constructive review. We will clarify the empirical grounding by foregrounding the three case studies, add clear signposts to the figure/timeline when they are discussed, and compress rhetoric in favor of concise claims tied to the case studies and Marr’s framework.

**3-3 Review Response2:**

ApcN

**3-4 Reaction To Review2:**

Point taken on scope. We will explicitly mark where prompting is limited (e.g., small or pre-linguistic models) and indicate contexts where mechanistic approaches are more appropriate.

**3-5 Review Response3:**

yFki

**3-6 Reaction To Review3:**

Thank you for your thoughtful review. We will clarify how the case studies and Marr’s framework provide concrete grounding for our position. We will also highlight how the paper addresses reproducibility through prompt protocols and scope boundaries. Finally, we will make the complementarity with mechanistic interpretability more apparent.

---

### Meta-Review · Area_Chair_8Z4Z · 2025-09-06

**Rating:** 7
**Confidence:** 3

**Strengths:**

– Clear, thorough, and persuasive arguments.
– Deep discussions regarding why prompting is undervalued.
– Thoughtfully addresses common critics of prompting.
– Adequate discussion of alternative positions.
– Bridges conceptual and practical divides.

**Weaknesses:**

– Position mostly argued through rhetoric.
– Lack of discussion regarding why observations from prompting are hard to formalize.
– Lack of discussion regarding “casual vs. serious” prompting, making parallels with other disciplines.
– Position assumes models trained on natural language.
– Lack of empirical support.

**Questions:**

– omDN’s question regarding other fields.
– ApcN’s question regarding models without command of natural language.
– yFki’s questions regarding frameworks, interpretability, and evolving models.

**Ethics:**

No.

**Thoroughness:**

3

---

### Decision · Program_Chairs · 2025-09-26

Accept